# Repeated sleep disruption in mice leads to persistent shifts in the fecal microbiome and metabolome

Samuel J. Bowers[1,2]*, Fernando Vargas[3], Antonio González[4], Shannon He[1,2], Peng Jiang[1,2], Pieter C. Dorrestein[3,5], Rob Knight[4,5,6], Kenneth P. Wright, Jr[7,8,9], Christopher A. Lowry[7,8], Monika Fleshner[7,8], Martha H. Vitaterna[1,2], Fred W. Turek[1,2,10,11]

1 Center for Sleep and Circadian Biology, Northwestern University, Evanston, Illinois, United States of America, 2 Department of Neurobiology, Northwestern University, Evanston, Illinois, United States of America, 3 Collaborative Mass Spectrometry Innovation Center, Skaggs School of Pharmacy & Pharmaceutical Sciences, University of California San Diego, La Jolla, California, United States of America, 4 Department of Pediatrics, University of California San Diego School of Medicine, La Jolla, California, United States of America, 5 Center for Microbiome Innovation, University of California San Diego, La Jolla, California, United States of America, 6 Department of Computer Science and Engineering, University of California San Diego, La Jolla, California, United States of America, 7 Department of Integrative Physiology, University of Colorado Boulder, Boulder, Colorado, United States of America, 8 Center for Neuroscience, University of Colorado Boulder, Boulder, Colorado, United States of America, 9 Sleep and Chronobiology Laboratory, University of Colorado Boulder, Boulder, Colorado, United States of America, 10 The Ken & Ruth Davee Department of Neurology, Northwestern University Feinberg School of Medicine, Chicago, Illinois, United States of America, 11 Department of Psychiatry and Behavioral Sciences, Northwestern University Feinberg School of Medicine, Chicago, Illinois, United States of America

* samuel.bowers@northwestern.edu

**Data Availability Statement:** Sequencing data and metadata are available on Qiita under study ID 10777 and on EBI-ENA with accession number EBI: ERP113564. The metabolomics dataset is

## Abstract

It has been established in recent years that the gut microbiome plays a role in health and disease, potentially via alterations in metabolites that influence host physiology. Although sleep disruption and gut dysbiosis have been associated with many of the same diseases, studies investigating the gut microbiome in the context of sleep disruption have yielded inconsistent results, and have not assessed the fecal metabolome. We exposed mice to five days of sleep disruption followed by four days of *ad libitum* recovery sleep, and assessed the fecal microbiome and fecal metabolome at multiple timepoints using 16S rRNA gene amplicons and untargeted LC-MS/MS mass spectrometry. We found global shifts in both the microbiome and metabolome in the sleep-disrupted group on the second day of recovery sleep, when most sleep parameters had recovered to baseline levels. We observed an increase in the *Firmicutes*:*Bacteroidetes* ratio, along with decreases in the genus *Lactobacillus*, phylum *Actinobacteria*, and genus *Bifidobacterium* in sleep-disrupted mice compared to control mice. The latter two taxa remained low at the fourth day post-sleep disruption. We also identified multiple classes of fecal metabolites that were differentially abundant in sleep-disrupted mice, some of which are physiologically relevant and commonly influenced by the microbiome. This included bile acids, and inference of microbial functional gene content suggested reduced levels of the microbial bile salt hydrolase gene in sleep-disrupted mice. Overall, this study adds to the evidence base linking disrupted sleep to the gut microbiome and expands it to the fecal metabolome, identifying sleep disruption-sensitive

publicly available in the MassIVE database under accession number MSV000080630 (https://massive.ucsd.edu/ProteoSAFe/dataset.jsp?task=8f3141b17a1e4b5886df0d4c515f2a16).

**Funding:** This research was funded by the Office of Naval Research Grant # N00014-15-1-2809 (SJB, FV, AG, SH, PJ, PCD, RK, KPW, CAL, MF, MHV, FWT), https://www.onr.navy.mil/. Also with support from National Institutes of Health Training Grant T32HL007909 (SJB, MHV, FWT) https://researchtraining.nih.gov/programs/training-grants. The funders had no role in study design, data collection and analysis, decision to publish, or preparation of the manuscript.

**Competing interests:** The authors have declared that no competing interests exist.

bacterial taxa and classes of metabolites that may serve as therapeutic targets to improve health after poor sleep.

## Introduction

Inadequate sleep can lead to metabolic[1], immunologic[2, 3], and cognitive deficits[4]. Many of the pathological states that arise from sleep disruption also occur in conjunction with gut dysbiosis, defined as a disruption of the community structure of the gut microbiome. This includes metabolic disease[5–7] and cognitive impairment[8, 9] as well as other proinflammatory and neuro-behavioral disorders such as multiple sclerosis[10], depression[11], anxiety[9, 12], and posttraumatic stress disorder[13]. This has led to the hypothesis that there is a relationship between inadequate sleep and the gut microbiome. Only a small number of studies have tested this hypothesis, using heterogeneous sleep disruption protocols (e.g., acute sleep restriction[14, 15], chronic sleep fragmentation[16]), in humans[14, 17], mice[15, 16], and rats [17], and have yielded mixed results[17]. More research is therefore required to explore the relationship between sleep, the gut microbiome, and potential mediators of microbe-host interactions.

Despite the mounting evidence that supports an important role for the gut microbiome in normal physiology, the mechanisms by which commensal microorganisms influence the host are still unclear. Proposed mechanisms include direct interactions with the enteric nervous system[18], interactions with toll-like receptors in the intestinal epithelium[19], regulation of the immune system[20], and signaling of microbially-modified metabolites including those originating from food sources and host bile acids[21–26]. These metabolites serve as a functional measure of microbial activity, and the fecal metabolome closely reflects the composition of the fecal microbiome[27]. Therefore, to understand the impacts of the microbiome on the host, it is crucial to study not only the microbes, but also to examine the molecules that they produce and that are present in their microenvironment. However, there have been no studies to date examining the effects of sleep disruption on the fecal metabolome using untargeted metabolomics.

We thus investigated the impact of a sub-chronic, five-day sleep disruption protocol on the fecal microbiome and fecal metabolome in mice. Assessment of the fecal microbiome using 16S rRNA gene amplicons and of the fecal metabolome using untargeted LC-MS/MS mass spectrometry revealed a global shift in both the microbiome and metabolome after sleep disruption, and aspects of these changes persisted through the fourth day after returning to *ad libitum* sleep. Furthermore, microbial differential abundance testing and utilization of Global Natural Products Social Molecular Networking[28] (GNPS) allowed us to identify specific taxa of bacteria and families of metabolites that change in response to five days of sleep disruption, many of which have known physiological relevance. These findings support the hypothesis that gut dysbiosis, and changes in the fecal metabolome, after sleep disruption may contribute to some of the health problems long known to be associated with inadequate sleep and that these changes may be present even after the sleep-wake state is normalized.

## Materials and methods

### Animals

Seven-week old male C57BL/6N mice (Experiment 1, $N = 7$; Experiment 2, $N = 20$; Charles River Laboratories, USA) were used in these experiments. Mice were group-housed upon

arrival for one week until surgery (Experiment 1) or until being placed into individual sleep disruption chambers (Experiment 2). Mice were maintained on a 12:12 L:D cycle at room temperature (23°C ± 2°C) with food and water available *ad libitum* throughout the experiment. The light source was two 14 W fluorescent tubes (soft white, 3000 K), resulting in an average light intensity of ~500 lux inside the cylindrical sleep disruption cage. Zeitgeber Time (ZT) is defined as the number of hours after the onset of the light period (light onset = ZT0). All mice were housed and handled according to the Federal Animal Welfare guidelines, and all studies were approved in advance by the Institutional Animal Care and Use Committee at Northwestern University (Assurance Number: A3283-01; Protocol Number IS00001718).

## Sleep disruption protocol

The same sleep disruption protocol was used in Experiment 1 and Experiment 2. Prior to the sleep disruption protocol, all mice were transferred from their home cages into individual sleep disruption cylindrical cages. Cages had corncob bedding and food/water available *ad libitum*. Mice were allowed to acclimate to the chambers for 7 days before beginning the sleep disruption protocol. Sleep disruption was achieved using a commercially available system integrated into the chambers (Pinnacle Technology, Lawrence, KS, USA), which simulates the gentle handling technique via a rotating metal bar (22 cm in length) attached to a post at the center of the cage. For the sleep disruption period, the bar's rotation speed was set at seven rotations per minute with reversals of rotation direction (i.e., clockwise vs. counterclockwise) set to occur at semi-random intervals of 10 ± 10 seconds. The bar was programmed to rotate for 20 hours per day (ZT6-ZT2), and was stationary from ZT2-ZT6, for 5 days. Experimenters visually inspected mice at regular intervals during the sleep disruption windows to ensure that the bar mechanism was functioning properly and that the sleep-disrupted mice were awake. Control animals were placed in identical cages with bars that remained stationary throughout the experiment. At ZT2 of the fifth sleep disruption day, the motorized bars were stopped, and mice were allowed to sleep *ad libitum* for the remainder of the experiment.

## Sleep recording and analysis

One week after arrival, mice for Experiment 1 were implanted with electroencephalographic/ electromyographic (EEG/EMG) sleep recording devices (Pinnacle Technologies, Lawrence, KS, USA). Surgical procedures were performed using a mouse stereotaxic apparatus with standard aseptic techniques in a ventilated, specially-equipped surgical suite. Anesthesia was induced by IP injection of cocktail of ketamine HCl (98 mg/kg; Vedco Inc, St. Joseph, MO, USA) and xylazine (10 mg/kg; Akorn Inc, Lake Forest, IL, USA) before surgical implantation of a headmount, which consisted of a plastic 6-pin connector connected to four EEG electrodes and two EMG electrodes. Four stainless steel screws serving as anchors for the EEG leads and grounds were screwed into the skull with one screw located 1 mm anterior to bregma and 2 mm lateral to the central suture, and the other at 1 mm anterior to lambda and 2.5 mm lateral to the central suture. The exposed ends of two stainless steel Teflon-coated wires (0.002 in. in diameter) serving as EMG leads were then inserted into the nuchal muscles using a pair of forceps. The headmount was then sealed by dental acrylic and a single suture at the front and back of the implant was given to close the skin. Subcutaneous injection of analgesic meloxicam (2 mg/kg; Norbrook Laboratories, Newry, Northern Ireland) was given to the animals immediately after the surgery while the animals were still under anesthesia and once more on the following day.

One week after implant surgery, mice were moved into cylindrical sleep recording cages (25 cm in diameter and 20 cm tall, Pinnacle Technologies) within individual acoustically isolated chambers and the headmount was connected to the transmission tether. Cages had

corncob bedding and food/water available *ad libitum*. Two days were allowed for acclimation to the tether before baseline sleep was recorded. Sleep was recorded continuously throughout the remainder of the experiment. Data were collected using Pinnacle Acquisition software (Pinnacle Technologies), then scored as non-rapid eye movement sleep (NREM), rapid eye movement sleep (REM), or Wake in 10 second epochs using machine learning-assisted sleep scoring software developed in the Turek/Vitaterna laboratory[29]. The initiation of a bout of NREM, REM, or Wake was defined by the occurrence of two consecutive epochs of NREM, REM, or Wake (respectively). A bout was terminated when a bout of another state occurred. Sleep bouts were initiated by two consecutive epochs of a sleep state (NREM or REM) and were only terminated when a wake bout occurred. The delta power band was defined as 0.5–4 Hz. Relative power was calculated as the raw power (uV^2) in a particular band divided by the total power in all bands. Power was then reported as a percent of baseline to reduce inter-individual variability.

## Fecal sample collection

In Experiment 2, fecal samples were collected at 3 different timepoints: 1) after mice were transferred to sleep disruption cages but before starting sleep disruption (BL); 2) on the second afternoon (~30 h) after the sleep disruption protocol was ended (R2); and 3) on the fourth afternoon after the sleep disruption protocol was ended (R4). At each collection, mice were placed into a clean sleep disruption chamber with fresh bedding and food and monitored closely until two fresh fecal pellets from each mouse were collected. Only spontaneously voided pellets were collected, so not every animal produced fecal pellets at every timepoint. Samples were placed into individual 1.5 mL microfuge tubes, and frozen at -80˚C until microbiome/ metabolome analysis. All fecal pellets were collected between ZT8 and ZT12.

## Microbiome analysis

Microbiome data were generally analyzed using the Quantitative Insights Into Microbial Ecology 2 (QIIME2, version 2018.2) bioinformatics software package[30, 31]. A total of 56 fecal samples (BL: *n* = 10/10 Control/Sleep Disruption; R2: *n* = 8/8; R4: *n* = 10/10) were processed for microbiome analyses. DNA was extracted from fecal samples and the V4 region of the 16S rRNA gene was amplified using the 515f/806rB primer pair with the barcode on the forward read[32] and sequenced as previously described[33] using an Illumina MiSeq. Sequence data were processed using Deblur v1.0.2[34], trimming to 150 nucleotides to create sub-operational-taxonomic-units (sOTUs). These were then inserted into the Greengenes 13_8[35] 99% reference tree using SATe-enabled Phylogenetic Placement (SEPP)[36]. SEPP uses a simultaneous alignment and tree estimation strategy[37] to identify placements for sequence fragments within an existing phylogeny and alignment. Taxonomy was assigned using an implementation of the Ribosomal Database Project (RDP) classifier[38] as implemented in QIIME2[30].

The OTU feature table was filtered to remove any features present in three or fewer samples (out of the 56 original samples), and alpha and beta diversity metrics were performed at a rarefied depth of 8431 reads, resulting in the removal of five samples from the dataset (final *n* for diversity metrics: Control/Sleep disruption—BL: *n* = 8/8; R2: *n* = 8/8; R4: *n* = 10/9). Beta diversity was assessed using weighted UniFrac distance[39] matrices, which were used to generate PCoA plots and to perform PERMANOVA in QIIME2. Within-group distance was calculated from distance matrices by averaging the weighted UniFrac distance from an individual sample to all other samples in the same group (Control vs Sleep Disruption) at the same timepoint. Distance from baseline was calculated by averaging the distance from an individual sample at R2 or R4 to all samples in the same group at the BL timepoint. Alpha diversity metrics were

calculated using scikit-bio 0.5.1 as implemented by QIIME2. To test for differentially abundant taxa between control and sleep-disrupted groups, samples with less than 8000 reads were removed (final *n* for differential abundance testing: Control/Sleep Disruption—BL: *n* = 8/8; R2: *n* = 8/8; R4: *n* = 10/9), and DESeq2 (version 1.14.1) was performed on the non-rarefied dataset at each timepoint and at each taxonomic level using the Bioconductor R package in RStudio (version 1.0.136, RStudio Inc). This was used in favor of techniques that more adequately account for the compositionality of microbiome datasets[40] such as Analysis of the Composition of Microbiomes (ANCOM) due to the extremely low sensitivity of ANCOM when sample size is less than 20 per group[41].

## PICRUSt2 analysis of 16S rRNA gene data

We inferred the microbial gene content from the taxa abundance using the software package Phylogenetic Investigation of Communities by Reconstruction of Unobserved States (PICRUSt2; https://github.com/picrust/picrust2; v2.1.4-b)[42]. This tool allows assessment of functional capacity of a microbiome using 16S rRNA sequencing data. We then used DESeq2 to identify genes that were differentially abundant between control and sleep-disrupted groups (notated with Enzyme Commission numbers).

## Metabolomic analysis

A total of 56 fecal samples (BL: *n* = 10/10 Control/Sleep Disruption; R2: *n* = 8/8; R4: *n* = 10/10) were processed for analysis of the fecal metabolome. Fecal samples were analyzed using an ultra-high performance liquid chromatography system coupled to a quadrupole-Orbitrap mass spectrometer (Q Exactive, Thermo Scientific, Waltham, MA, USA). Chromatographic separation was accomplished using a Kinetex C18 1.7 μm, 100 Å pore size, 2.1 mm (internal diameter) x 50 mm (length) column (Phenomenx, Torrance, CA, USA). The column was maintained at 40˚C during chromatographic separation. 5.0 μL of extract was injected per sample. Mobile phase composition was (A) water with 0.1% formic acid (v/v) and (B) acetonitrile with 0.1% formic acid (v/v) with a flow rate of 0.5 mL/min. Chromatographic elution was performed as follows: 0.00–0.50 min, 5% B; 0.50–4.00 min, 50% B; 4.00–5.00 min, 99% B; 5.00–7.00 min, 99% B; 7.00–7.10 min, 5% B; 7.10–9.00 min, 5% B. Positive mode electrospray ionization was performed using a heated electrospray ionization source using the following source parameters: spray voltage, 3500 V; capillary temperature, 268.75˚C; sheath gas flow rate, 52.50 (arb. units); auxiliary gas flow rate, 13.75 (arb. units); probe heater temperature, 437.50˚C; and S-lens RF level, 50 (arb. units). Mass spectrometry data were collected using data-dependent acquisition. The MS1 scan range was set to 150–1,500 *m/z* with a resolution of 17,500 at 200 *m/z*. MS2 scans of the five most abundant ions in the previous MS1 scan, acquired in a data-dependent manner, were collected at a resolution of 17,500 at 200 *m/z*. MS1/MS2 automatic gain control target and maximum ion injection time were set to 5.0 E5 and 100 ms respectively. Higher-energy collision-induced dissociation was performed with a normalized collision energy stepped from 20, 30, to 40%.

The LC/MS/MS feature table, generated using Optimus[43] peak detection, was normalized to an internal standard followed by a row sum (total ion count) normalization and filtered to remove features present in less than two samples. The resulting table contained 1124 metabolites. PCoA plots were then generated using Bray-Curtis distance, and PERMANOVA was performed at each timepoint on the normalized feature table using the Vegan package (version 2.5–5) in RStudio. In order to identify metabolites that were different between sleep-disrupted and control groups, we used a multiple-method approach that included machine learning and nonparametric hypothesis testing. In order to first identify the group of metabolites that were

the key drivers of differences between groups at each timepoint, Variable Selection Using Random Forests (VSURF, version 1.0.3)[44, 45] analysis was performed using the VSURF.R package in RStudio. Briefly, this protocol uses multiple iterations of the random forest supervised machine learning technique to isolate the most important drivers of separation between two groups by defining a threshold variable importance. Taking this list of suprathreshold features, we then performed Wilcoxon Rank Sum tests at each timepoint as a form of a 'post hoc' test to confirm differences between groups.

Features of interest were annotated using GNPS (version 1.3.0) [28], which allows MS1 and MS2 spectra to be shared between researchers, forming a large database. By matching an unknown spectrum to one or more in the database, and examining similarity to others within a molecular network, GNPS can be used to identify purported molecular structures of features from untargeted metabolomics. A molecular network was created using the online workflow at GNPS. The data were filtered by removing all MS/MS peaks within +/- 17 Da of the precursor $m/z$. MS/MS spectra were window-filtered by choosing only the top six peaks in the +/- 50 Da window throughout the spectrum. The data were then clustered with MS-Cluster with a parent mass tolerance of 0.1 Da and a MS/MS fragment ion tolerance of 0.1 Da to create consensus spectra. Further, consensus spectra that contained less than two spectra were discarded. A network was then created where edges were filtered to have a cosine score above 0.6 and more than four matched peaks. Further edges between two nodes were kept in the network if and only if each of the nodes appeared in each other's respective top ten most similar nodes. The spectra in the network were then searched against GNPS' spectral libraries. The library spectra were filtered in the same manner as the input data. All matches kept between network spectra and library spectra were required to have a score above 0.6 and at least four matched peaks. Results can be found at https://gnps.ucsd.edu/ProteoSAFe/status.jsp?task= 6fb1d63a51764c7ea75a4e7256b6936a

Individual features of interest from the feature table were then matched to nodes (clusters) in the network whose average $m/z$ and RT were within 0.025 and 30 s, respectively, of the feature of interest. Features that were matched to multiple clusters using the aforementioned criteria were assigned to the cluster with the closest average $m/z$ and RT.

## Statistical analysis and software

All graphs depict the mean +/- SEM unless otherwise stated. All PCoA plots were generated using the EMPeror visualization tool as implemented in QIIME2[46]. Microbiome data processing and analysis, including PERMANOVA, were performed in QIIME2 as outlined above. Wilcoxon Rank-Sum tests, Kruskal-Wallace tests, VSURF, DESeq2 (with Benjamini Hochberg adjustment), heatmaps, and boxplots/scatterplots were performed or generated in RStudio (version 1.0.136, RStudio Inc). Two way ANOVA and mixed-effects models with Bonferroni post hoc testing of sleep, alpha diversity, and beta diversity measures, along with generation of all other graphs/figures, was performed using GraphPad PRISM (version 8.2.1; GraphPad Inc, San Diego, CA, USA). Test statistics generated by PERMANOVA, ANOVA, and mixed-effects models are reported in **S1 Table**.

## Results

### The five-day sleep disruption protocol significantly reduces and fragments sleep

In Experiment 1, we performed a detailed analysis of sleep before, during, and after the sleep disruption protocol that was used in Experiment 2 (see **Methods** and **Fig 1A**). Compared to

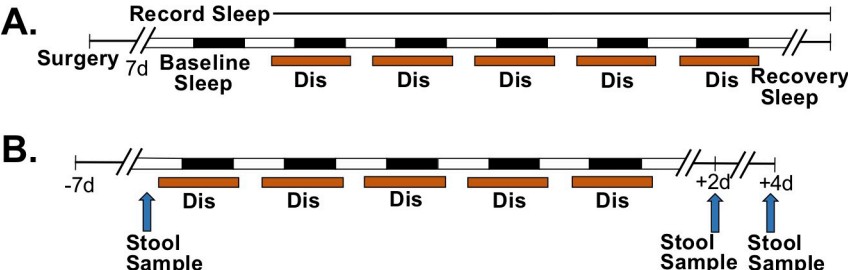

**Fig 1. Experimental timelines.** A) Experiment 1. Adult male C57BL/6N mice (*n* = 3, Control; *n* = 4, Sleep Disruption) received surgical implants of sleep recording devices. After recovery from surgery, mice were subjected to five days of repeated sleep disruption and two days of *ad libitum* recovery sleep. Sleep was disrupted for 20 h/day, with an *ad libitum* sleep window between ZT2-ZT6. Sleep was recorded throughout the experiment. B) Experiment 2. Non-instrumented adult male C57BL/6N mice (*n* = 10/group) were subjected to the same sleep disruption protocol, but with four days of recovery sleep. Stool samples were collected before sleep disruption, on day 2 post-sleep disruption and on day 4 post-sleep disruption (arrows). Abbreviations: Dis, Sleep Disruption.

control animals, sleep-disrupted animals had significantly less total sleep, NREM sleep, and REM sleep per 24 hours during the protocol (**Fig 2A–2C**, F statistics can be found in **S1 Table**). While the amount of 24-hour NREM sleep recovered to the level of controls within the first day of recovery sleep (**Fig 2B**), there was a significant rebound in the amount of REM sleep on the first day of recovery sleep (**Fig 2C**). In order to observe sleep with greater resolution, we examined the fifth day of the sleep disruption protocol and the beginning of the first day of recovery sleep using two-hour time bins. It was evident that the majority of this 24-hour sleep loss occurred during the hours of the light period in which the motorized sleep disruption bar was moving (ZT0-ZT2 and ZT6-ZT12, **Fig 2E–2G**). REM sleep was reduced to nearly zero percent while the motorized bar was moving, and this resulted in strong REM rebounds during the first two hours of the *ad libitum* recovery windows (**Fig 2G**). Sleep disruption also resulted in more fragmented sleep. During the five days of sleep disruption, there was a significantly higher number of state changes in sleep-disrupted mice compared to controls (**Fig 2D and 2H**). Furthermore, there was an increase in the number of bouts of sleep and bouts of NREM in the sleep disruption group, accompanied by a significant decrease in the bout length, further suggesting fragmentation (**S1 Fig**). The number of REM bouts per 24 hours was significantly decreased on days the motorized bar was on, was significantly increased on the first recovery day, but no longer significantly different from control by the second recovery day (**S1 Fig**).

## Five days of sleep disruption creates changes in the fecal microbiome that last at least four days after disruption has ended

In Experiment 2, fecal samples were collected before sleep disruption (BL), at day two post-sleep disruption (R2), and at day four post-sleep disruption (R4) (see **Methods** and **Fig 1B**) to assess the fecal microbiome and fecal metabolome. Beta diversity, or the difference in diversity between two or more communities, was assessed at each experimental time point with weighted UniFrac distance, which takes into account both the abundances and phylogenetic relatedness of two communities[39, 50]. Principal coordinates analysis (PCoA) revealed no difference between control and sleep-disrupted groups at baseline, as expected (*p* = 0.877, PERMANOVA; **Fig 3A**), but significant clustering of control mice and sleep-disrupted mice indicated a global difference in community structure at R2 (*p* = 0.018, PERMANOVA; **Fig 3B**) that was gone by R4 (*p* = 0.663, PERMANOVA; **Fig 3C**). The distance from baseline, the average weighted UniFrac distance between an individual post-sleep disruption and all individuals

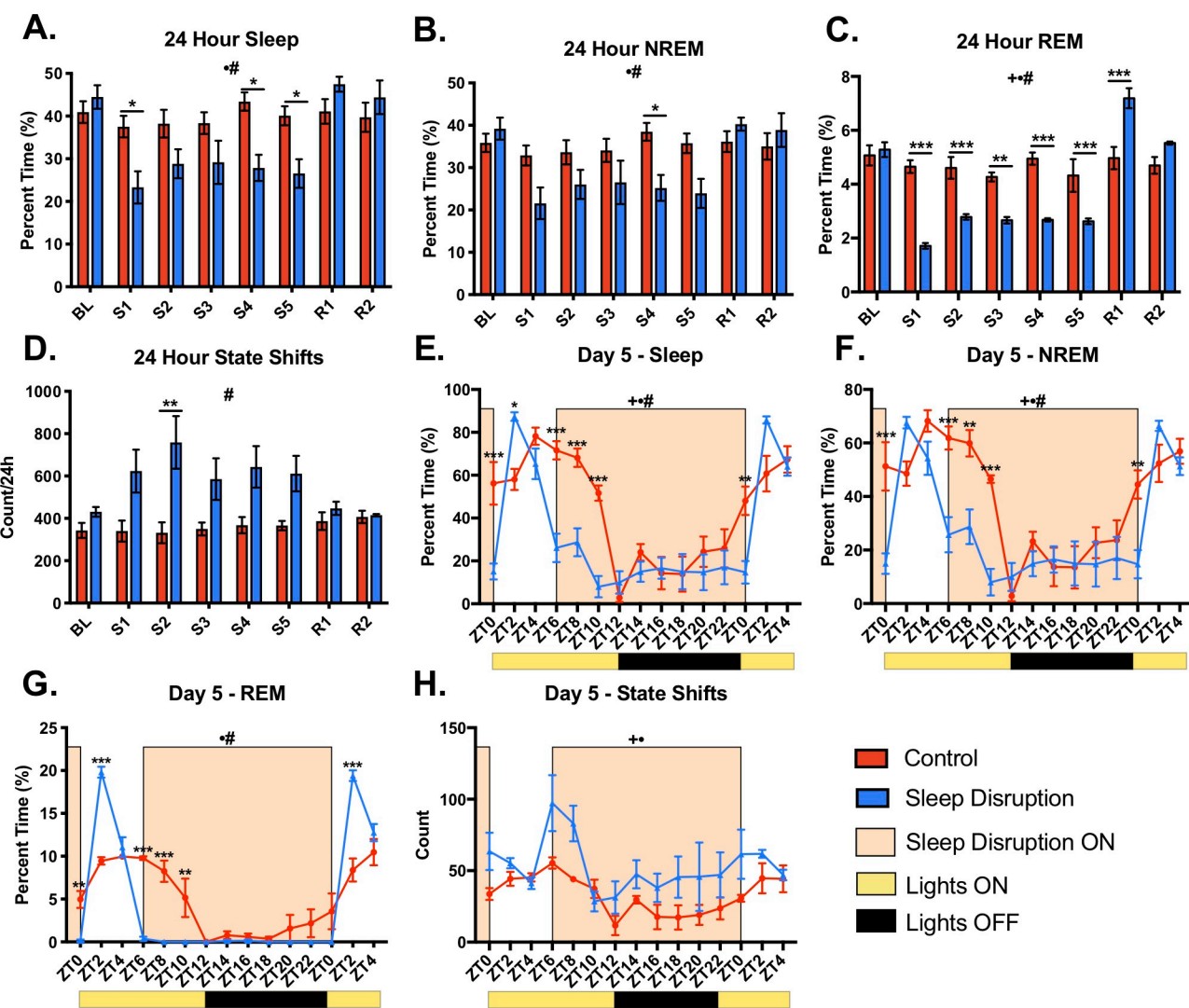

**Fig 2. Effect of sleep disruption protocol on sleep measures.** A-D) 24-hour totals of total sleep, non-rapid eye movement sleep (NREM), rapid eye movement sleep (REM), and state shifts. There was a significant decrease in sleep, NREM, and REM during the sleep disruption protocol, and an increase in state shifts. E-H) Two-hour bins of total sleep, NREM, REM, and state shifts from the fifth day of the sleep disruption protocol through ZT4 of the first day of recovery sleep. Yellow bars under the x axis indicate the lights being on, while black bars indicate the lights being off. Abbreviations: BL, baseline; S, sleep disruption; R, recovery; ZT, zeitgeber time. $n$ = 3-4/group. $^{*}p < 0.05$, $^{**}p < 0.01$, $^{***}p < 0.001$ (Bonferroni post hoc test); $+p < 0.05$ (overall effect of sleep disruption over entire time interval, Mixed-effects model); $•p < 0.05$ (overall effect of Time over entire time interval, Mixed-effects model); $\#p < 0.05$ (Sleep DisruptionxTime interaction over entire time interval, Mixed-effects model).

within the same group at BL, was increased at R2 and R4 (**Fig 3D, Right panel**). Furthermore, sleep disruption significantly increased the dissimilarity between individuals within the sleep-disrupted group at R2 and R4 (**Fig 3D, Left panel**). Therefore, five days of repeated sleep disruption had a "destabilizing" effect in that it not only shifted microbial communities away from controls, it increased dissimilarity within the group, and this effect lasted at least four days after return to *ad libitum* recovery sleep.

Multiple measures of alpha diversity, the microbial diversity within an individual community, were also examined because reductions in alpha diversity have been associated with pathological states such as inflammatory bowel syndrome[51], chronic stress[52], and obesity[53]. Faith's phylogenetic diversity index was not affected by sleep disruption (**Fig 3E, Left**),

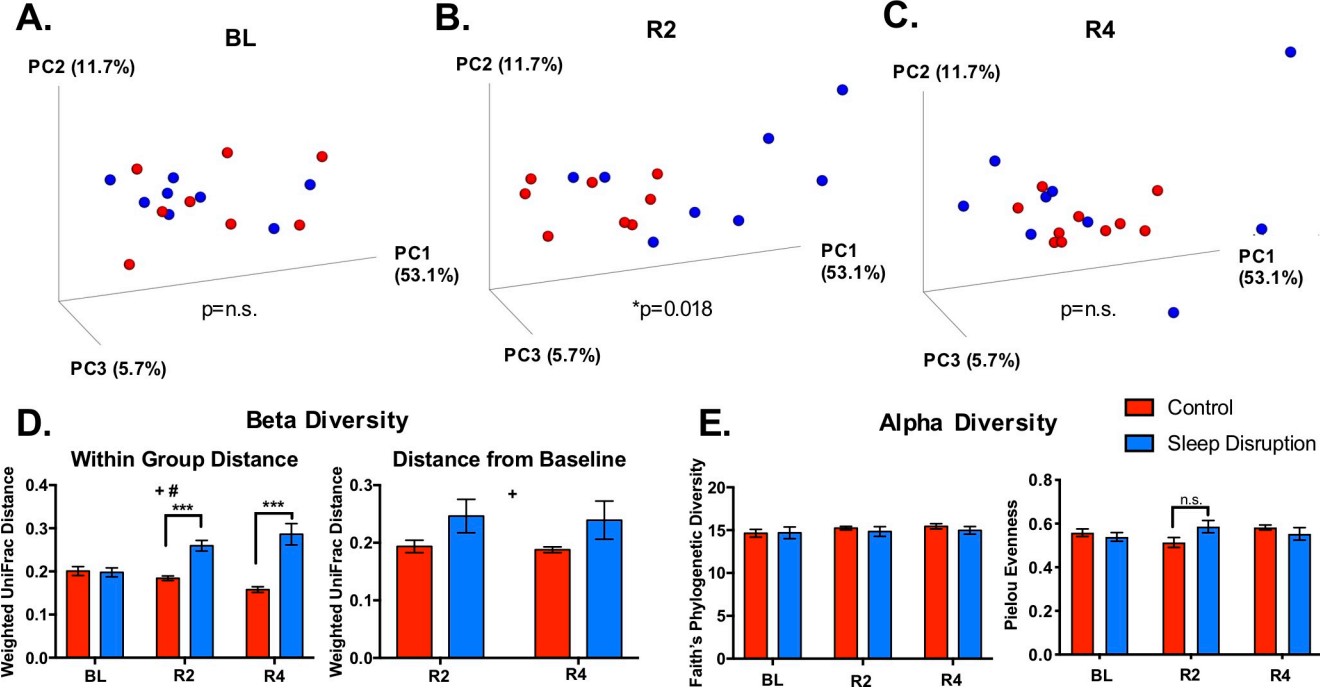

**Fig 3. Effect of sleep disruption on microbiome beta and alpha diversity.** A-C) Principal coordinates analysis (PCoA) plots using weighted UniFrac distance. A significant difference between sleep disruption and control groups at day 2 post-sleep disruption was detected using PERMANOVA. D) Average weighted UniFrac distance from an individual to all individuals within the same group (left) and from an individual post-sleep disruption to each individual pre-sleep disruption (right) is increased at both day 2 and day 4 post-sleep disruption. E) Faith's Phylogenetic Diversity (left) and Pielou Evenness (right) were unchanged throughout the experiment. Abbreviations: BL, baseline; R2, day 2 post-sleep disruption; R4, day 4 post-sleep disruption. $n$ = 8-10/group.
$^{*}p < 0.05$ (PERMANOVA); $^{**}p < 0.01$, $^{***}p < 0.001$ (Bonferroni post hoc test); $+p < 0.05$ (Overall effect of Sleep Disruption, Mixed-effects model); $\#p < 0.05$ (Sleep Disruption x Time interaction, Mixed-effects model).

consistent with results in different sleep disruption models[16]. Pielou evenness[54] was also unaffected by sleep disruption (**Fig 3E, Right**).

## Multiple bacterial taxa are differentially abundant in the sleep-disrupted group

We tested for differential abundance between control and sleep disruption groups at each taxonomic level, at each timepoint. Of the 142 originally identified taxa (includes all levels), 0, 16, and 6 were significantly different at the BL, R2, and R4 timepoints, respectively (FDR < 0.1, **Fig 4A**; **S2 Table**). The ratio of the two most prevalent phyla in the mammalian gut, the *Firmicutes:Bacteroidetes* (F:B) ratio, is a blunt measure of community shift. An increase in the F:B ratio has been seen in obesity[55, 56], stress[57], as well as models of acute[14] and chronic [16] sleep disruption. We found a significant sleep disruption-induced increase in the F:B ratio (**Fig 4B**) that was significant at R2 but not at R4.

The increase in the F:B ratio was due to a significant increase in the relative abundance of *Firmicutes* at R2 (**Fig 4C**). Within the *Firmicutes* phylum, two major classes changed in different directions at R2. *Bacilli* were significantly decreased in sleep-disrupted mice (**Fig 4D**), while *Clostridia* were significantly increased (**Fig 4G**). The decrease in the class *Bacilli* appeared to mostly be due to significant decreases in the genus *Lactobacillus* (**Fig 4E**) and genus *Turicibacter* (**Fig 4F**). Within the *Clostridia* class, one unknown genus within the *Clostridiaceae* family was significantly decreased at R2 (**Fig 4H**), while other taxa within class *Clostridia* were significantly increased (**S2 Table**). The low abundance phylum *Actinobacteria* (**Fig**

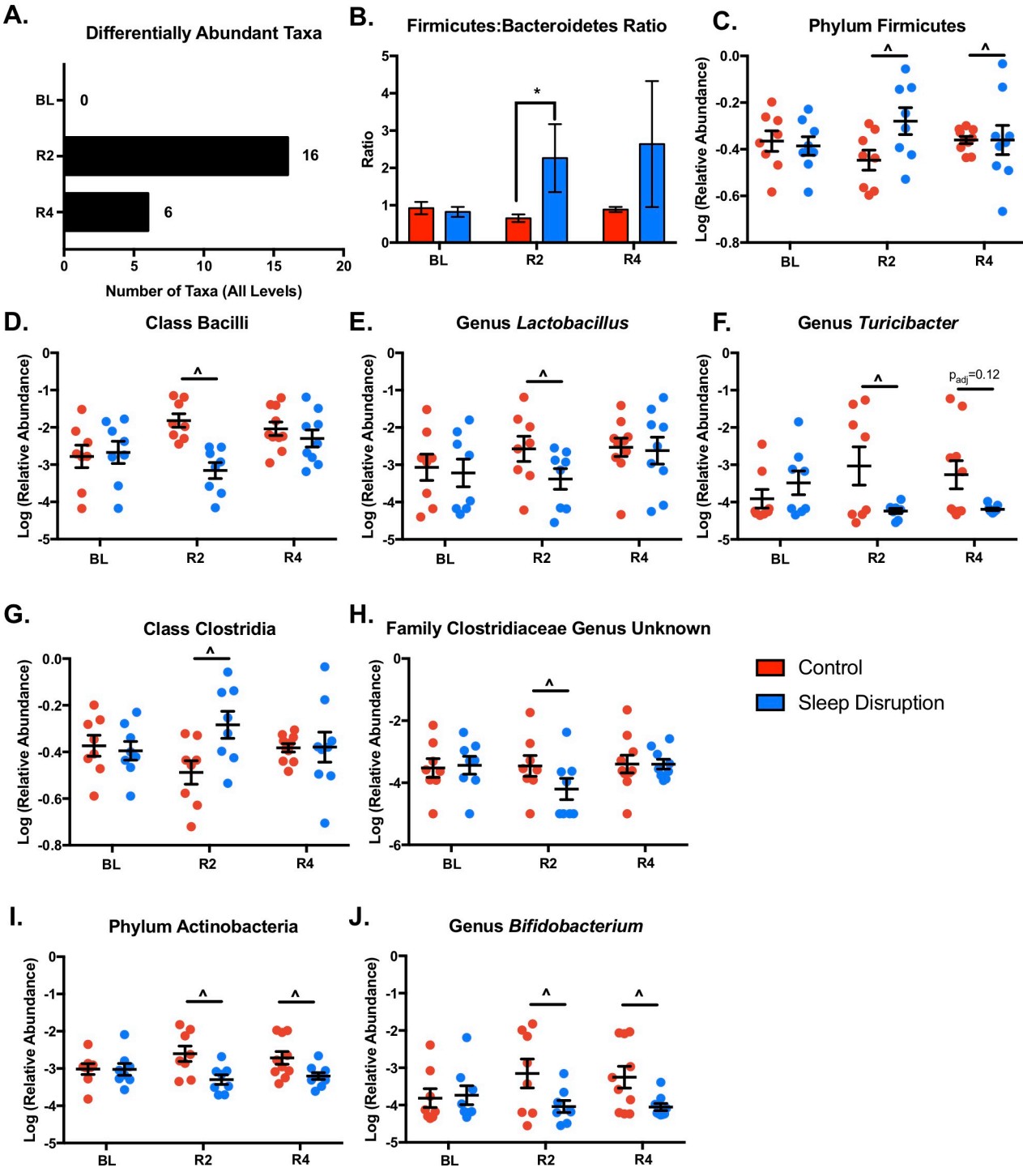

**Fig 4. Effect of sleep disruption on individual microbial taxa.** At each timepoint, DESeq2 was performed at each taxonomic level to identify taxa differentially abundant between sleep disrupted and control groups. A) Summary of significantly different taxa by timepoint. B) The ratio of relative abundances of the phyla *Firmicutes* to *Bacteroidetes* was significantly increased at day 2 post-sleep disruption in sleep-disrupted animals. This increase was mostly driven by a significant increase in *Firmicutes* (C). Within the *Firmicutes* phylum, the class *Bacilli* (D), genus *Lactobacillus* (E), and genus *Turicibacter* (F) were reduced at day 2 post-sleep disruption. The class *Clostridia* was increased (G) and an unknown genus within the *Clostridiaceae* family (H) was decreased in sleep-disrupted animals at day 2 post-sleep disruption. Both the phylum *Actinobacteria* (I), and the genus *Bifidobacterium* (J) were decreased at day 2 and day 4 post-sleep disruption in the sleep disrupted group. Abbreviations: BL, baseline; R2, day 2 post-sleep disruption; R4, day 4 post-sleep disruption. $n$ = 8-10/group. Data represent means ± SEM. *$p < 0.05$ (Wilcoxon Rank-Sum Test); ^FDR < 0.1 (DESeq2).

**4I**) was significantly decreased in the sleep-disrupted group at both R2 and R4. This decrease was evident in the genus *Bifidobacterium* within the *Actinobacteria* phylum. These results parallel the beta diversity findings in that the greatest magnitude of shift in the fecal microbiome was at R2, and while some measures recover, others persist into R4.

### Five days of sleep disruption changes the fecal metabolome

Due to the increasing evidence supporting the role microbes play in generating or altering physiologically active metabolites, we examined the impact of sleep disruption on the fecal metabolome. Normalized feature tables containing 1124 features were used for PCoA analysis at each timepoint to assess global changes due to sleep disruption. No separation was observed at BL ($p = 0.881$, PERMANOVA; **Fig 5A**), but a clear separation between sleep-disrupted and control mice was seen at R2 ($p = 0.007$, PERMANOVA; **Fig 5B**). This separation was no longer present at R4 ($p = 0.381$, PERMANOVA; **Fig 5C**). Of the 1124 molecular features assayed, 250 were identified as significantly changing over time, relative to BL, in either the control group, the sleep-disrupted group, or both (Kruskal-Wallace FDR < 0.1, **Fig 5D**). Many features (101/250) significantly changed only in control animals, suggesting sleep disruption prevented a naturally occurring change. Conversely, 57/250 features significantly changed over time in the sleep-disrupted mice but not in the non-sleep-disrupted mice. We also compared sleep disruption to control groups at each timepoint individually to assess the relative amount of differentially abundant features at each stage of the experiment, and the majority of significantly differentially abundant features (142/204; Wilcoxon Rank-Sum, $p < 0.05$) were found at R2 (**Fig 5E**), with 57 of those 142 decreased and 85 of the 142 increased in the sleep-disrupted group. Only 20/204 (13 decreased, 7 increased in the sleep disruption group) significantly differentially abundant features were found at BL, whereas 42/204 (29 decreased, 13 increased in the sleep disruption group) significantly differentially abundant features were found at R4. Overall, these results indicate that five days of sleep disruption results in a global shift in the fecal metabolome, both preventing naturally occurring shifts in the abundances of some metabolites and creating changes in others. While this global shift is present only at R2, some metabolites remained altered on the 4$^{th}$ day of *ad libitum* recovery sleep.

### A subset of metabolites drive separation between sleep-disrupted and control groups at day two post-sleep disruption

Variable Selection Using Random Forests (VSURF)[44, 45] was used to identify features that were important drivers of separation between sleep disruption and control groups at R2. VSURF identified 98 features that were above the threshold variable importance (suprathreshold) and that successfully distinguished the two groups on a heatmap (**Fig 6A and 6B**; **S3 Table**).

From here we sought to learn about the possible identities of these features of interest using GNPS[28]. By matching an unknown spectrum or cluster of spectra to spectra in a large database, and examining their similarity to others within a molecular network, GNPS can be used to identify molecular classes and annotate purported molecular structures of features from untargeted metabolomic datasets. This is a level 2 or 3 metabolite identification according to the 2007 Metabolomics Standards Initiative[58], where level 1 is considered a high confidence identification. Using GNPS to generate a molecular network for this dataset, the MS2 spectra of 21/98 suprathreshold R2 features were matched to annotated spectra, including 4 of the top 25 drivers identified by VSURF (**Fig 6B**). Examining only the top annotated features (**Fig 6C**), many features with spectral matches to di- and tripeptides, along with the lysine degradation product L-saccharopine, were significantly increased in the sleep-disrupted group (**Fig 6D–6F**; **S2 Fig**).

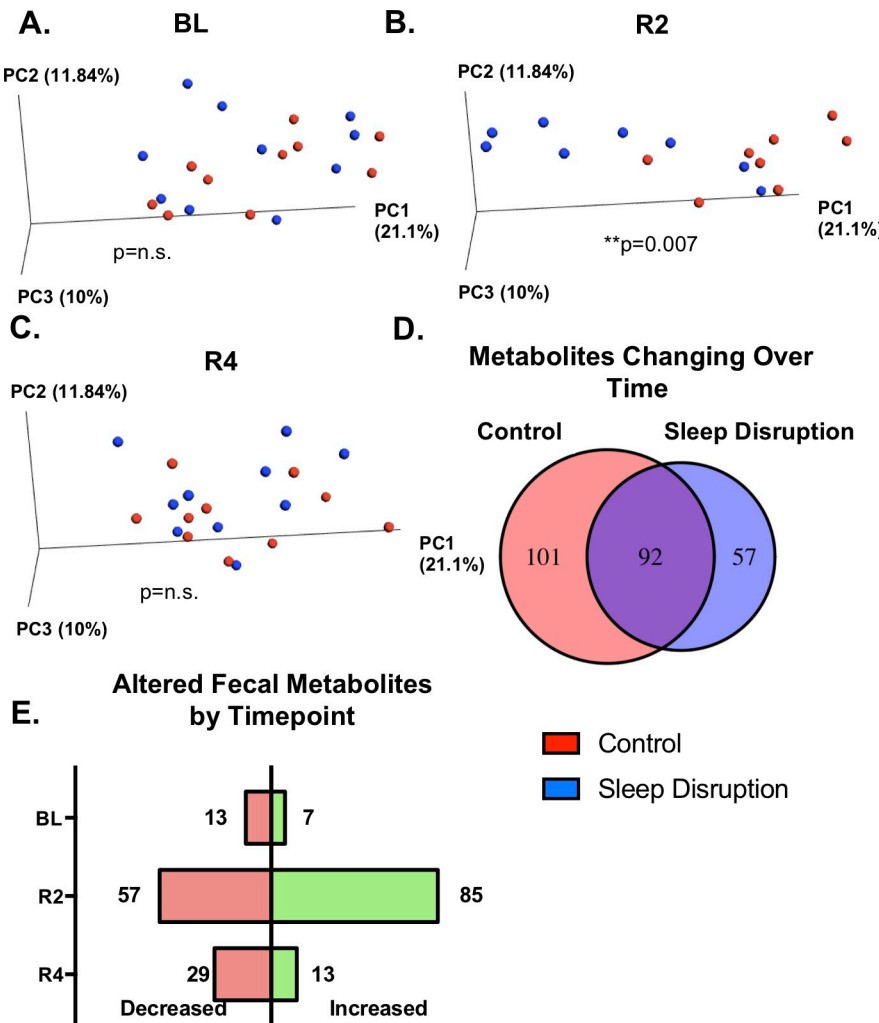

**Fig 5. Effect of sleep disruption on the fecal metabolome.** A,B,C) Principal coordinates analysis (PCoA) plots using Bray Curtis distance. PERMANOVA detected a significant difference between sleep disruption and control groups day 2 post-sleep disruption, but not BL or at day 4 post-sleep disruption. D) Kruskal-Wallace tests were run within the control group and within the sleep disruption group to determine metabolites significantly changing over the course of the experiment (FDR < 0.1). The number of metabolites found to have an effect of time only in the control group (left number), an effect of time only in the sleep-disrupted group (right number), or in both groups (middle number) is depicted in the Venn diagram. E) Wilcoxon Rank-Sum tests were performed at each timepoint to quantify the number of metabolites increased (right, green bars) or decreased (left, pink bars) in the sleep disruption group at each timepoint (uncorrected $p < 0.05$). Abbreviations: BL, baseline; R2, day 2 post-sleep disruption; R4, day 4 post-sleep disruption. $n = 8$-10/group. $^{**}p < 0.01$ (PERMANOVA).

We also noticed sleep disruption-induced changes in features with spectral matches to bacterially modified molecules including bile acids and urobilin. Two suprathreshold features with spectral matches to the bile acid cholic acid were significantly reduced, and two were significantly increased in sleep-disrupted mice at R2 (**Fig 6G and 6H**). Furthermore, two unannotated features with structural similarity to bile acids, as indicated by their presence in the same molecular networks as primary and secondary bile acids, were also significantly reduced (**S3 Fig**), indicating structural similarity to bile acids. Bile acids are commonly modified by bacteria in the gut lumen by the enzyme encoded by the gene *bile salt hydrolase* (BSH)[59, 60], and have diverse signaling properties that involve the immune[61] and nervous systems[25,

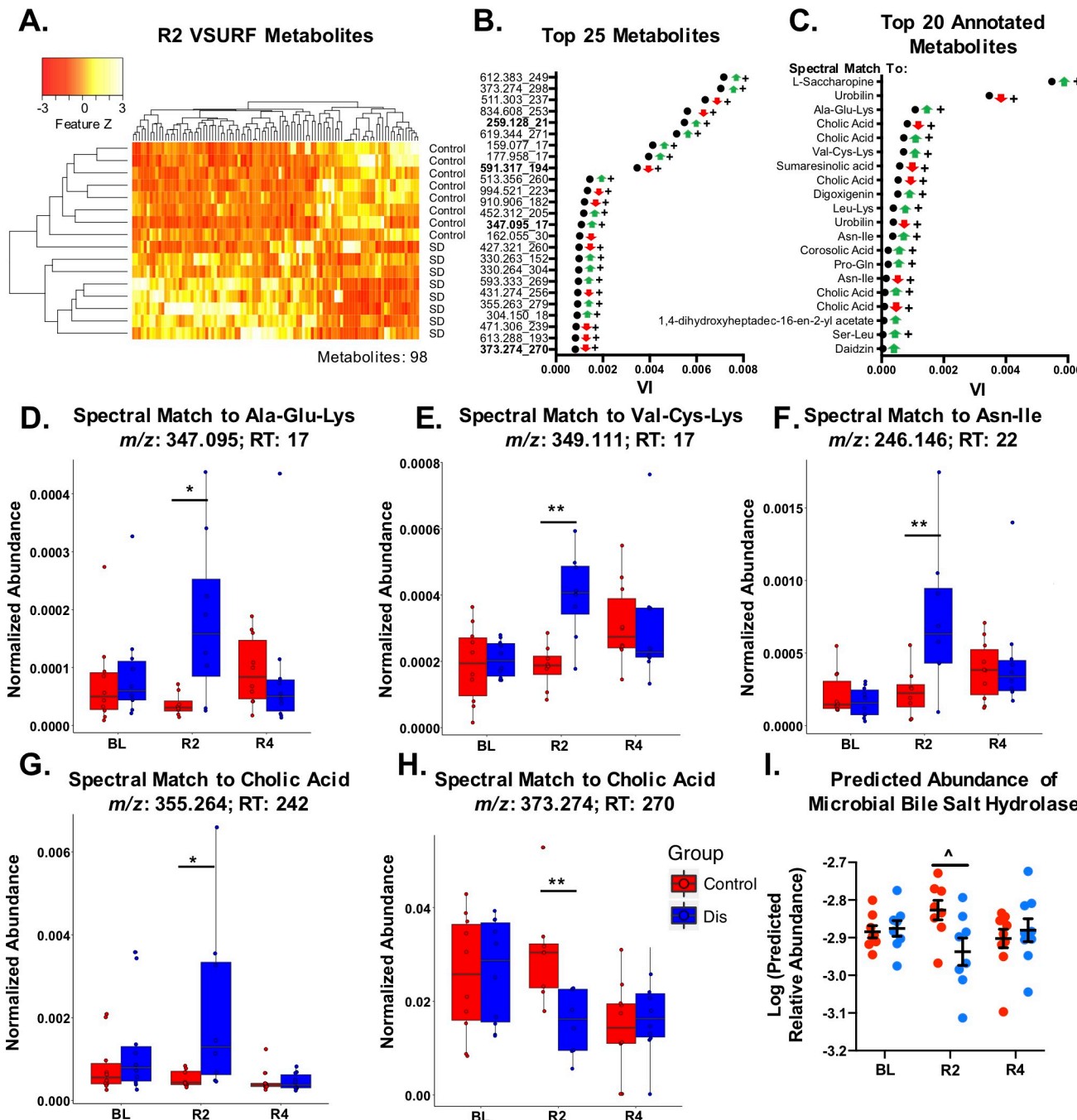

**Fig 6. Metabolites that drive changes seen at day 2 post-sleep disruption.** Variable Selection Using Random Forests (VSURF) was performed at the second day post-sleep disruption (R2) to identify metabolites that are the most important drivers (above a threshold variable importance) of separation between sleep-disrupted and control groups. A) Heatmap of the 98 suprathreshold metabolites. B) Variable importance scores of the top 25 suprathreshold metabolites ($m/z$_RT), along with direction of change (arrows, green/up = increased in sleep disruption group, red/down = decreased in sleep disruption group). Bold indicates metabolites that were annotated using Global Natural Products Social Molecular Networking (GNPS). C) Top 20 annotated metabolites. Normalized abundance (peak intensity normalized to total ion count) of metabolites with spectral matches to D) Ala-Glu-Lys, E) Val-Cys-Lys, and F) Asn-Ile were increased at R2 but not at day R4. G,H) One metabolite with a spectral match to cholic acid was increased (G) and another was decreased (H) at R2. I) Inferred abundance of the microbial bile salt hydrolase gene (EC:3.5.1.24) was also reduced at R2. For (D-H), boxes indicate median, 25th and 75th quantiles; whiskers indicate 2*IQR from edges of box. For (I), data represent mean ± SEM. Abbreviations: BL, baseline; R2, day 2 post-sleep disruption; R4, day 4 post-sleep disruption; Dis, Sleep Disruption; VI, variable importance; $m/z$, mass to charge ratio; RT, retention time (seconds). $n$ = 8-10/group. +FDR < 0.1, *$p$ < 0.05, **$p$ < 0.01 (Wilcoxon-Rank Sum test); ^FDR < 0.1 (DESeq2).

62]. Thus, we used the software package PICRUSt2 to infer microbial gene content from the 16S rRNA gene data and assess inferred abundance of microbial BSH in our fecal samples. We found the inferred abundance of BSH (EC:3.5.1.24) was significantly reduced in the sleep-disrupted group at R2 (**Fig 6I**, **S4 Table**). These results provide evidence that microbially modified, physiologically active classes of molecules are impacted by five days of sleep disruption, and that the microbiome has an altered functional capacity to produce them.

Another class of molecules that was affected by sleep disruption was dietary-derived pentacyclic triterpenoids. Triterpenoids are plant-derived molecules, and some have been shown to have anti-inflammatory properties[63]. We identified a molecular network containing 43 clusters, 12 of which were annotated as pentacyclic triterpenoids or close derivatives (**S4 Fig**). Of the 43 clusters in the network, 9 were matched to VSURF suprathreshold features. This includes seven that were suprathreshold at R2 (**S4B–S4H Fig**) and two that were suprathreshold at R4 (**S4I and S4J Fig**). A feature with a spectral match to sumaresinolic acid (**S4B Fig**), along with unannotated feature 645 (**S4G Fig**) were significantly reduced at R2. A feature matching corosolic acid (**S4C Fig**), along with unannotated features with the ID's 871, 204, 133 and 273 (**S4D–S4F** **and S4H Fig**) were significantly increased at R2.

Another molecular network of interest contained two unannotated ions that only appeared in sleep-disrupted groups (**S5 Fig**). These metabolites therefore hold potential to act as markers for recent sleep disruption, and future additions to the GNPS database may result in level 2 or 3 annotation of clusters in the network.

## Some changes to fecal metabolites are present at day four post-sleep disruption

Although no global change was seen on PCoA, we also ran VSURF analysis on the R4 feature table and identified 64 suprathreshold metabolites that were able to separate the control and sleep-disrupted groups (**S6A and S6B Fig**; **S3 Table**). Seven of these metabolites had MS2 spectra that matched reference spectral libraries in GNPS. Among the annotated features were molecules with spectral matches to hederagenin and wilforlide A (**S4I and S4J Fig**), which were significantly reduced compared to controls at R4, and fell into the same molecular network as multiple metabolites that were suprathreshold at R2. Others that were significantly increased or decreased at R2 compared to controls did not recover by R4 (e.g. **S6D–S6F Fig**). This indicates that, while no global changes were evident by day four of recovery sleep, sleep disruption did have an impact on some individual metabolites that persisted for at least four days.

## Discussion

Repeated sleep disruption is ubiquitous in modern society and has been linked to a multitude of health problems. Recent lines of scientific inquiry have established an important role for the gut microbiome in multiple facets of mammalian health and disease, many of which are also affected by sleep disruption. The present study took a detailed look at the impact of a subchronic, five-day sleep disruption protocol on the fecal microbiome as well as the fecal metabolome in mice, and found that repeated exposure to inadequate sleep had an impact on the microbiome and metabolome that lasted at least four days after the sleep disruption had ended. Importantly, sleep disruption reduced levels of beneficial bacterial genera, altered the metabolic function of the microbiome, and changed fecal levels of bacterially modified metabolites such as bile acids. These results can provide insights into possible mechanisms by which sleep disruption may impact host physiology.

The protocol used in this study resulted in severely disrupted sleep for five days, characterized by an decrease in sleep amounts, and an increase in fragmentation. This pattern was fairly

stable across the five days, indicating that the animals were unable to adapt to the paradigm or find strategies to improve sleep as the protocol went on. This is a relevant model because repeated nights of inadequate sleep followed by a few days of recovery sleep is a common schedule in society today, and short sleep mixed with fragmentation is particularly prominent among groups with demanding work schedules such as on-call physicians[64] and active duty military personnel[65]. By the second day of recovery sleep, nearly all sleep parameters had returned to control levels. Whether the specific characteristics of sleep disruption determine how the microbiome or metabolome changes is unknown and warrants further investigation.

Previous work has demonstrated chronic (four week) sleep fragmentation (short disruption every two minutes) in mice alters the gut microbiome[16]. Our results suggest that a sub-chronic, more severe sleep disruption paradigm also results in significant shifts in the microbial community structure, without large variation in alpha diversity. This sleep disruption protocol also increased the dissimilarity of the fecal microbiome between animals within the sleep disruption group (within group distance) at both R2 and R4, indicating a "destabilizing" effect that persisted long into recovery sleep. The sleep-wake pattern had normalized by R2, suggesting that recovery of the microbiome may be a slower process than sleep homeostatic mechanisms.

Changes to particular taxa observed in sleep-disrupted mice suggest the dysbiosis induced by repeated inadequate sleep may have a detrimental physiological impact. Differential abundance analysis of individual taxa revealed multiple bacterial taxa that were increased or decreased in the sleep-disrupted group compared to controls, including an increase in the *Firmicutes*:*Bacteroidetes* (F:B) ratio and a decrease in *Lactobacillus*, *Actinobacteria*, and *Bifidobacterium*, all of which have established physiological impacts. An increase in the F:B ratio is a blunt measure of community shift and has been seen in many pathological states including obesity[55, 56], chronic stress[57], as well as an acute short sleep paradigm in humans[14] and a chronic sleep fragmentation paradigm in rodents [16]. The phylum *Actinobacteria*, genus *Bifidobacterium*, and genus *Lactobacillus* were all low in sleep-disrupted mice. Previous studies in rodents and humans support a positive role for these taxa in stress resilience[13, 66] and anxiety-like behavior[67–69]. Therefore, an increased F:B ratio, along with reduced *Actinobacteria*, *Bifidobacterium*, and *Lactobacillus* indicates a state whereby ability to cope with a proinflammatory or anxiety-inducing stimulus may be reduced. Indeed, sleep deprivation results in altered responsiveness of the hypothalamic-pituitary-adrenal (HPA) axis[70–72], increased inflammation[73, 74], and potentiated effects of a chemical stressor in a model of colitis in mice[75], indicating that sleep deprivation may be a factor promoting stress vulnerability. This is supported by a human study that observed that preexisting complaints of poor sleep increased the risk of posttraumatic stress disorder (PTSD) and other stress-related psychiatric disorders following trauma exposure[76].

As one of the principle mechanisms by which a change in the fecal microbiome can impact host physiology is via change in the molecules they produce, we also examined the fecal metabolome in this study. An untargeted metabolomic approach allows for wide surveillance of the molecular environment as well as discovery of new molecular classes of interest[77]. Although untargeted mass spectrometry cannot confirm exact structures of metabolites of interest without secondary targeted standard assays, GNPS[28] allows us to infer the general class of many interesting features based on spectral matches and molecular networking. Using this approach, we identified multiple classes of molecules significantly impacted by sleep disruption, including bile acids, which are microbially modified and physiologically relevant.

Multiple results from this study suggest microbiome-influenced bile acid metabolism was impacted by sleep disruption. Primary bile acids are cholesterol derivatives that are synthesized in the mammalian liver and excreted into the intestinal lumen to aid in lipid emulsification

and absorption. In the intestine, primary bile acids are dehydroxylated and deconjugated by the gut microbiome, creating secondary bile acids and greatly enhancing the diversity of the bile acid pool[78]. Mounting evidence in the past decade has described bile acids as versatile signaling molecules, with receptors throughout the mammalian organism[26, 62, 78]. Some bile acids can act as anti-inflammatory and immunoregulatory agents in the intestinal tract and the central nervous system by activating the bile acid receptors FXR (farnesoid X receptor) and TGR5 (Takeda G protein-coupled receptor 5)[25, 61]. Furthermore, bile acid receptors play a role in glucose, lipid, and cholesterol metabolism[79, 80]. Two molecules with spectral matches to cholic acid and two unannotated molecules within molecular networks that contained multiple primary and secondary bile acids were decreased in the sleep-disrupted group at R2.

Importantly, analysis of the inferred gene content also revealed a reduction in the abundance of the microbial *bile salt hydrolase* (BSH) gene in the microbiomes of sleep-disrupted mice at R2. BSH catalyzes the critical first step in microbial bile acid metabolism, and multiple lines of evidence suggest these enzymes may be the "gatekeepers" of host-microbiome crosstalk[60]. In a previous experiment, feeding *Escherichia coli* engineered to overexpress *Lactobacillus* BSH to mice protected them from weight gain, and curbed lipid and cholesterol metabolism[81]. A reduction in the fecal bile acid pool due to a reduction in microbial BSH, therefore, could result in a proinflammatory, metabolically dysregulated state in the host. Indeed, increased inflammatory markers have been observed in a sub-chronic short sleep (10 nights of 4 hours of sleep opportunity per night) model in humans [82], while a similar protocol (6 nights of 4 hours of sleep opportunity per night; also in humans) reduced glucose tolerance and increased sympathetic nervous system activity [1].

We also noticed a general increase in the abundance of metabolites with spectral matches to tripeptides and dipeptides. This could indicate an increase in host mucosal proteolysis or in microbial proteolysis[83]. Host and microbial proteolytic enzymes play a role in gastrointestinal physiology, including activating signaling pathways (e.g. protease-activated receptors or PAR's) controlling inflammation and gut barrier function[83] as well as modulation of dorsal root ganglion neuron excitability[84]. Furthermore, commensal bacteria have been shown to create molecules with potent protease inhibitory activity[85], so a shift in microbial community structure could have a direct impact on host proteolysis and physiology.

A third class of molecules that was impacted by sleep disruption was pentacyclic triterpenoids and close derivatives. The molecular family of dihydroxylated pentacyclic triterpenoids, judged by spectral matches to sumaresinolic acid and corosolic acid, were decreased and increased, respectively, at R2, while the dihydroxylated and monohydroxylated spectral features, with spectral matches to hederagenin and wilforlide A, were significantly decreased at R4. Also, four unannotated spectra within the network were significantly increased, and one was decreased, at R2. Triterpenoids are a class of diverse, plant-derived molecules that have been traditionally studied for their anti-tumor or anti-inflammatory properties[63, 86, 87]. Shifts in the balance of this molecular network could therefore have impacts on host physiology. While it has been shown that administration of triterpenoid molecules can modulate the microbiome[88], and that certain pentacyclic triterpenoids are metabolized by the microbiome[89], it is unclear whether the changes seen in this family of molecules due to sleep disruption were due to changes in the microbiome. In order to evaluate the biological impacts and therapeutic potential of the molecules discovered in this study, follow-up studies will need to be done to verify the structures of the features discovered here as well as to quantify their concentrations in the gut.

Our results are consistent with a study by Poroyko *et al*. investigating the gut microbiome in a mouse model of obstructive sleep apnea[16]. In that study, four weeks of chronic sleep fragmentation caused global shifts in the fecal microbiome, as well as an increase in the F:B

ratio, similar to this study. Furthermore, a study of acute sleep loss (four-hour sleep opportunity) in humans also observed an increase in the F:B ratio, but not a global shift in beta diversity[14]. Recently, a study in rats found marked shifts in the fecal microbiome and urinary metabolites after a seven day severe stress/REM deprivation protocol[90]. Taken with the present results, a link between inadequate sleep and the fecal microbiome appears to be present across species and sleep disruption protocols. Importantly, the present study expands on these findings to include the fecal metabolome, which has important implications as an effector system of microbe-host interactions[21].

It is of note that a study published by Zhang et al.[17] used a similar sleep restriction protocol (20h/day sleep disruption using a rotating bar for 7 days) but found no changes in the fecal microbiome. There are a few potential reasons for this discrepancy. First, the study by Zhang and colleagues used rats while ours used mice. Second, the rats in both the sleep restriction and control groups were manipulated every day to collect body weight measurements and fecal pellets. We chose to leave the animals relatively undisturbed throughout the sleep disruption protocol. Our automated protocol allowed mice to remain in the same home cage to have undisturbed sleep opportunity and to have limited contact with human researchers, which can affect the microbiome[91, 92]. However, this approach did introduce some limitations to the experiment because it did not allow for constant monitoring of food intake or fecal microbiome/metabolome during the sleep disruption period.

Overall, this study characterizes the impact of inadequate sleep on fecal microbiome as well as the fecal metabolome, a potential effector system in microbe-host interactions. The changes to microbiome and metabolome were present on the second day of recovery sleep, and some changes persisted until at least the fourth day of recovery sleep, despite the recovery of most sleep within two days of the cessation of sleep disruption. This is particularly interesting in view of the observation that some of the neurobehavioral impairments observed after a week of short sleep do not recover after a 'weekend' of recovery sleep despite reduction of subjective sleepiness[93, 94]. Our findings also suggest that changes seen in particular bacteria and bacterially-influenced signaling molecules such as bile acids suggest a proinflammatory, metabolically dysregulated state in the days following a five-day sleep disruption protocol. Interventions designed to maintain the fecal microbiome and metabolome, or to proactively offset the negative impacts of dysbiosis, should be investigated to promote resilience to repeated sleep disruption, a problem that is ubiquitous in modern society.

## Supporting information

**S1 Fig. Effect of sleep disruption protocol on sleep fragmentation measures and delta power.** A,B) There was a significant increase in the 24-hour totals of total sleep bouts (A) and non-rapid eye movement sleep (NREM) bouts (B) during the sleep disruption protocol in the sleep disruption group. C) Rapid eye movement (REM) bouts were decreased during the sleep disruption protocol, and increased on the first day of recovery in the sleep-disrupted group. D, E,F) Median sleep bout duration (D) and NREM bout duration (E) were significantly decreased in the sleep-disrupted group during the protocol, while the median REM bout duration (F) was unaffected in all days except for on S5. G) There was no change in 24-hour NREM delta power due to sleep disruption. Abbreviations: BL, baseline; S, sleep disruption; R, recovery; ZT, zeitgeber time. $n$ = 3-4/group. $^*p < 0.05$, $^{**}p < 0.01$, $^{***}p < 0.001$ (Bonferroni post hoc test); $+p < 0.05$ (overall effect of sleep disruption, Mixed-effects model); $\bullet p < 0.05$ (overall effect of Time, Mixed-effects model); $\#p < 0.05$ (Sleep DisruptionxTime interaction, Mixed-effects model).
(PDF)

**S2 Fig. Sleep disruption changes fecal levels of molecules related to protein metabolism.** A) Normalized abundance (peak intensity normalized to total ion count) of a metabolite with a spectral match to Leu-Lys was increased at day 2 post-sleep disruption. B) Global Natural Products Social Molecular Networking (GNPS)-generated molecular network containing multiple annotated (blue) and unannotated (grey) clusters. Numbers next to grey clusters indicate average parent mass of the spectra in the cluster. Metabolites that were above the threshold variable importance in Variable Selection Using Random Forests (VSURF) analysis are outlined with a black circle. Length of grey lines connecting clusters indicates relative similarity of the MS2 spectra. Arrowheads point towards clusters with a larger $m/z$. C) Normalized abundance of a metabolite with a spectral match to L-saccharopine was increased at day 2 post-sleep disruption (R2). Boxes indicate median, 25$^{th}$ and 75$^{th}$ quantiles; whiskers indicate 2*IQR from edges of box. Abbreviations: BL, baseline; R2, day 2 post-sleep disruption; R4, day 4 post-sleep disruption; Dis, Sleep Disruption; VI, variable importance; $m/z$, mass to charge ratio; RT, retention time (seconds). $n = 8$-$10$/group. *$p < 0.05$, **$p < 0.01$ (Wilcoxon-Rank Sum test). (PDF)

**S3 Fig. Sleep disruption decreases fecal levels of unknown molecules in networks with bile acids.** A,D) Global Natural Products Social Molecular Networking (GNPS)-generated molecular networks containing multiple annotated (blue) and unannotated (grey) clusters. Numbers next to grey clusters indicate average parent mass of the spectra in the cluster. Metabolites that were above the threshold variable importance in Variable Selection Using Random Forests (VSURF) analysis are outlined with a black circle. Length of grey lines connecting clusters indicates relative similarity of the MS2 spectra. Arrowheads point towards clusters with a larger $m/z$. B) Normalized abundance (peak intensity normalized to total ion count) of an unannotated metabolite with ID 516 was decreased at day 2 post-sleep disruption (R2). C) Normalized abundance (peak intensity normalized to total ion count) of an unannotated metabolite with ID 512 was decreased at R2. Boxes indicate median, 25$^{th}$ and 75$^{th}$ quantiles; whiskers indicate 2*IQR from edges of box. Abbreviations: BL, baseline; R2, day 2 post-sleep disruption; R4, day 4 post-sleep disruption; Dis, Sleep Disruption; VI, variable importance; $m/z$, mass to charge ratio; RT, retention time (seconds). $n = 8$-$10$/group. *$p < 0.05$, **$p < 0.01$ (Wilcoxon-Rank Sum test). (PDF)

**S4 Fig. Sleep disruption changes fecal levels of molecules related to pentacyclic triterpenoids.** A) Global Natural Products Social Molecular Networking (GNPS)-generated molecular network containing multiple annotated (blue) and unannotated (grey) clusters. Numbers next to grey clusters indicate average parent mass of the spectra in the cluster. Metabolites that were above the threshold variable importance in Variable Selection Using Random Forests (VSURF) analysis are outlined with a black circle. Letters next to the black circles indicate the panel of the figure corresponding to the cluster. Length of grey lines connecting clusters indicates relative similarity of the MS2 spectra. Arrowheads point towards clusters with a larger $m/z$. B,C) Normalized abundance (peak intensity normalized to total ion count) of a metabolite with a spectral match to sumaresinolic acid (B) was decreased at day 2 post-sleep disruption, while a spectral match to corosolic acid (C) was increased at R2. D,E,F,H) Unannotated molecules with ID 871 (D), 204 (E), 133 (F), and 273 (H) were increased at R2. G,I,J) Unannotated molecule ID 645 (G) was decreased at R2. Metabolites matching hederagenin (I), and wilforlide A (J) were decreased at R4. Boxes indicate median, 25$^{th}$ and 75$^{th}$ quantiles; whiskers indicate points within 2*IQR from edges of box. Abbreviations: BL, baseline; R2, day 2 post-sleep disruption; R4, day 4 post-sleep disruption; Dis, Sleep Disruption; VI, variable importance; $m/z$, mass to charge ratio; RT, retention time (seconds). $n = 8$-$10$/group. *$p < 0.05$, **$p < 0.01$,

[***]$p < 0.001$ (Wilcoxon-Rank Sum test).
(PDF)

**S5 Fig. Two unknown fecal metabolites are present only in sleep-disrupted mice.** A) Global Natural Products Social Molecular Networking (GNPS)-generated molecular network containing multiple unannotated (grey) clusters. Numbers next to grey clusters indicate average parent mass of the spectra in the cluster. Metabolites that were above the threshold variable importance in Variable Selection Using Random Forests (VSURF) analysis are outlined with a black circle. Length of grey lines connecting clusters indicates relative similarity of the MS2 spectra. Arrowheads point towards clusters with a larger *m/z*. B) Normalized abundance (peak intensity normalized to total ion count) of an unannotated metabolite with ID 964 was increased at day 2 post-sleep disruption and at day 4 post-sleep disruption. C) Normalized abundance (peak intensity normalized to total ion count) of an unannotated metabolite with ID 965 was increased at day 2 post-sleep disruption but not day 4 post-sleep disruption. Boxes indicate median, 25[th] and 75[th] quantiles; whiskers indicate points within 2*IQR from edges of box. Abbreviations: BL, baseline; R2, day 2 post-sleep disruption; R4, day 4 post-sleep disruption; Dis, Sleep Disruption; VI, variable importance; *m/z*, mass to charge ratio; RT, retention time (seconds). $n = 8$-10/group. [*]$p < 0.05$, [**]$p < 0.01$ (Wilcoxon-Rank Sum test). (PDF)

**S6 Fig. Metabolites that are changed at day 4 post-sleep disruption.** A) Heatmap of the 64 metabolites that were above threshold variable importance in Variable Selection Using Random Forests (VSURF) analysis. B) Variable importance (VI) scores of the top 25 suprathreshold metabolites (*m/z*_RT). Bold indicates metabolites that were annotated using Global Natural Products Social Molecular Networking (GNPS). C) VI scores of annotated metabolites. D) Normalized abundance (peak intensity normalized to total ion count) of an unannotated metabolite with ID 241 was increased in sleep-disrupted relative to control mice at both day 2 post-sleep disruption (R2) and day 4 post-sleep disruption (R4). E) An unannotated metabolite with ID 661 was trending towards an increase in sleep-disrupted compared to control at R2 and was increased at R4. F) An unannotated metabolite with ID 155 was trending towards a decrease in sleep-disrupted mice compared to control mice at R2 and was decreased at R4. Boxes indicate median, 25[th] and 75[th] quantiles; whiskers indicate 2*IQR from edges of box. Abbreviations: BL, baseline; R2, day 2 post-sleep disruption; R4, day 4 post-sleep disruption; VI, variable importance; Dis, sleep disruption; *m/z*, mass to charge ratio; RT, retention time (seconds). $n = 8$-10/group. [*]$p < 0.05$, [**]$p < 0.01$ (Wilcoxon-Rank Sum test). (PDF)

**S1 Table. F statistics for PERMANOVA, ANOVA, and mixed-effects models.** Test statistics for analyses done in Fig 2, Fig 3, Fig 5 and S1 Fig are reported, and are organized by Figure number and panel letter. (XLSX)

**S2 Table. Differentially abundant bacterial taxa post-sleep disruption.** DESeq2 was performed at each taxonomic level to determine taxa differentially abundant between sleep disrupted and control groups. Taxa significant at an FDR < 0.1 are listed below as the mean relative abundance ± SEM, along with Benjamini Hochberg-adjusted *p* values. Fold difference: (Sleep Disruption-Control)/Control. Abbreviations: Dis, Sleep Disruption; Con, Control. $n = 8$-10/group. (DOCX)

**S3 Table. VSURF suprathreshold metabolites: Variable Selection Using Random Forests (VSURF) was performed at day 2 post-sleep disruption (R2) and day 4 post-sleep**

**disruption (R4) to identify top drivers of separation between sleep-disrupted and control groups.** Metabolites that were above the VSURF threshold variable importance ('suprathreshold') from each timepoint are listed above, along with the feature's ID, m/z ratio, retention time, annotated name (if any), fold difference in the sleep-disrupted group compared to the control group, unadjusted Wilcoxon Rank-Sum $p$ value, and Benjamini Hochberg (FDR)-adjusted $p$ value. Bold indicates adjusted $p$ values that are below the FDR of 0.1. Fold difference: (Sleep Disruption-Control)/Control. Abbreviations: Dis, sleep disruption; Con, control; MZ, mass to charge ratio; RT, retention time; N/A, not annotated; Inf, infinity. $n$ = 8-10/group.
(DOCX)

**S4 Table. PICRUSt2 results: PICRUSt2 software was used to infer microbial gene content from the 16S rRNA gene data, and DESeq2 was performed at pre-sleep disruption (BL), day 2 post-sleep disruption (R2), and day 4 post-sleep disruption (R4) to test for differentially abundant genes.** Results are listed above, with $p$-values, adjusted $p$-values, and fold-change ((Sleep Disruption-Control)/Control) listed for each detected Enzyme Commission number. Zero genes were significantly differentially abundant (FDR < 0.1) at BL, 176 were differentially abundant at R2, and zero were differentially abundant at R4. Abbreviations: Dis, sleep disruption; Con, control; N/A, not applicable; Inf, infinity. $n$ = 8-10/group.
(XLSX)

## Acknowledgments

The authors would like to acknowledge Dr. Gail Ackermann for organizing and coordinating data processing and analysis, and Chris Olker and Eun Joo Song for assistance scoring sleep.

## Author Contributions

**Conceptualization:** Samuel J. Bowers, Pieter C. Dorrestein, Rob Knight, Kenneth P. Wright, Jr, Christopher A. Lowry, Monika Fleshner, Martha H. Vitaterna, Fred W. Turek.

**Data curation:** Samuel J. Bowers, Fernando Vargas, Antonio González.

**Formal analysis:** Samuel J. Bowers, Fernando Vargas, Antonio González, Peng Jiang.

**Funding acquisition:** Pieter C. Dorrestein, Rob Knight, Kenneth P. Wright, Jr, Christopher A. Lowry, Monika Fleshner, Martha H. Vitaterna, Fred W. Turek.

**Investigation:** Samuel J. Bowers, Fernando Vargas, Antonio González, Shannon He, Peng Jiang.

**Methodology:** Antonio González.

**Project administration:** Pieter C. Dorrestein, Martha H. Vitaterna, Fred W. Turek.

**Resources:** Samuel J. Bowers, Fernando Vargas, Antonio González, Pieter C. Dorrestein, Rob Knight, Martha H. Vitaterna, Fred W. Turek.

**Software:** Samuel J. Bowers, Fernando Vargas, Antonio González, Peng Jiang.

**Supervision:** Pieter C. Dorrestein, Rob Knight, Kenneth P. Wright, Jr, Christopher A. Lowry, Monika Fleshner, Martha H. Vitaterna, Fred W. Turek.

**Visualization:** Samuel J. Bowers.

**Writing – original draft:** Samuel J. Bowers.

**Writing – review & editing:** Samuel J. Bowers, Fernando Vargas, Antonio González, Shannon He, Peng Jiang, Pieter C. Dorrestein, Rob Knight, Kenneth P. Wright, Jr, Christopher A. Lowry, Monika Fleshner, Martha H. Vitaterna, Fred W. Turek.

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
