## [Decision Letter · Decision Letter 0]

16 Jan 2020

PONE-D-19-35357

Repeated sleep disruption in mice leads to persistent shifts in the fecal microbiome and metabolome

PLOS ONE

Dear Mr. Bowers,

Thank you for submitting your manuscript to PLOS ONE. After careful consideration, we feel that it has merit but does not fully meet PLOS ONE’s publication criteria as it currently stands. Therefore, we invite you to submit a revised version of the manuscript that addresses the points raised during the review process.

We would appreciate receiving your revised manuscript by Mar 01 2020 11:59PM. To enhance the reproducibility of your results, we recommend that if applicable you deposit your laboratory protocols in protocols.io, where a protocol can be assigned its own identifier (DOI) such that it can be cited independently in the future. For instructions see: http://journals.plos.org/plosone/s/submission-guidelines#loc-laboratory-protocols

We look forward to receiving your revised manuscript.

Kind regards,

Juan J Loor

Academic Editor

PLOS ONE

Journal Requirements:

Reviewers' comments:

Reviewer's Responses to Questions

**Comments to the Author**

1. Is the manuscript technically sound, and do the data support the conclusions?

Reviewer #1: Yes

2. Has the statistical analysis been performed appropriately and rigorously? 

Reviewer #1: Yes

3. Have the authors made all data underlying the findings in their manuscript fully available?

Reviewer #1: Yes

4. Is the manuscript presented in an intelligible fashion and written in standard English?

Reviewer #1: Yes

5. Review Comments to the Author

Reviewer #1: The manuscript reported shifts in fecal microbiome and metabolome profiles in response to sub-chronic sleep disruption in mice. It is an interesting integration between microbial composition, function, and metabolites that would add new knowledge about the consequences of sleep disruption on gut health and functions, leading to impact the overall physiology in the host. The manuscript has a very detailed methodology and the manuscript is well written.

Throughout the manuscript: Stick to microbiome rather than microbiota

L595, L637: Provide the experimental model for these studies

6. PLOS authors have the option to publish the peer review history of their article (what does this mean?). If published, this will include your full peer review and any attached files.

Reviewer #1: No

---

## [Author Response · Author response to Decision Letter 0]

16 Jan 2020

Comments from Reviewer #1:

• “Throughout the manuscript: Stick to microbiome rather than microbiota”

o We have updated the terminology used in the manuscript accordingly. 

• “L595, L637: Provide the experimental model for these studies”

o We agree with the reviewer that it is important to describe the type of sleep disruption protocols used in the studies we cite at these locations in the paper. Therefore, we have amended and expanded the text to do so.

---

## [Editor Report · Decision Letter 1]

29 Jan 2020

Repeated sleep disruption in mice leads to persistent shifts in the fecal microbiome and metabolome

PONE-D-19-35357R1

Dear Dr. Bowers,

We are pleased to inform you that your manuscript has been judged scientifically suitable for publication and will be formally accepted for publication once it complies with all outstanding technical requirements.

With kind regards,

Juan J Loor

Academic Editor

PLOS ONE